# Cardiac Troponin I in Patients Undergoing Percutaneous and Surgical Myocardial Revascularization: Comparison of Analytical Methods

**DOI:** 10.3390/diagnostics13071316

**Published:** 2023-04-01

**Authors:** Sabrina Pacheco do Amaral Vendramini, Célia Maria Cássaro Strunz, Whady Armindo Hueb, Antonio de Padua Mansur

**Affiliations:** 1Laboratorio de Analises Clinicas, Instituto do Coracao (InCor), Hospital das Clinicas HCFMUSP, Faculdade de Medicina, Universidade de Sao Paulo, Sao Paulo 05403-900, SP, Brazil; 2Unidade Clinica de Aterosclerose, Instituto do Coracao (InCor), Hospital das Clinicas HCFMUSP, Faculdade de Medicina, Universidade de Sao Paulo, Sao Paulo 05403-900, SP, Brazil; 3Serviço de Prevencao, Cardiopatia na Mulher e Reabilitação Cardiovascular, Instituto do Coracao (InCor), Hospital das Clinicas HCFMUSP, Faculdade de Medicina, Universidade de Sao Paulo, Sao Paulo 05403-900, SP, Brazil

**Keywords:** troponin, CKMB, myocardial revascularization, biomarkers

## Abstract

The myocardial infarction (MI) types 4a and 5 guidelines recommend cardiac troponin (cTn) diagnostic decision limits of 5 and 10 times the 99th percentile, respectively. Different cTn kits elicit different responses, so the MI diagnosis is still challenging. The study aimed to establish the cutoff values and the accuracy of three different cTnI kits in the diagnosis of post-procedural MI. We analyzed 115 patients with multivessel stable chronic coronary artery disease; 26 underwent percutaneous coronary intervention, and 89 underwent coronary artery bypass graft. Delayed-enhancement magnetic resonance imaging was performed before and after each intervention for definitive MI diagnoses. Two contemporary and one high-sensitivity cTnI immunoassays were used. ROC curves determined the accuracy of each assay. Low accuracy was observed after applying the current guidelines recommendations. The three cTnI assays accuracies improved when adjusted by the new ROC cutoffs, reaching 82% for MI type 5 for all assays, and 78%, 88%, and 87% for MI type 4 for Siemens, Beckman, and Abbott, respectively. The ultrasensitive and contemporary tests’ accuracy for MI types 4a and 5 diagnoses are equivalent when adjusted for these new cutoffs. The hs-cTnI assays had lower accuracy than contemporary tests for MI types 4a and 5 diagnoses.

## 1. Introduction

Cardiovascular diseases (CVD) are currently the leading cause of death worldwide [1]. A recent study in Brazil, in a population aged 35 to 74 between 1996 and 2017, showed that mortality from CVD corresponded to 31% of all-cause deaths [2]. Coronary artery disease (CAD) is the CVD that most contributed to these statistics [3]. The primary etiopathogenic mechanism of CAD is the atherosclerotic disease. It is an immunoinflammatory disease [4], and stable angina, unstable angina, myocardial infarction (MI), and sudden death are the main clinical manifestations [5]. Lifestyle adjustments, pharmacological therapies, and invasive interventions change CAD’s natural history [6]. Coronary revascularization is the next step for highly symptomatic patients on optimal medical treatment. The percutaneous coronary intervention (PCI) or coronary artery bypass surgery (CABG) revascularization is effective in reducing symptoms, myocardial ischemia, and adverse cardiovascular events [7].

Of the 1,149,602 CAD interventions performed from 2008 to 2018, PCI accounted for 755,557 (66%) and CABG 244,105 (21%) procedures. The PCI/CABG ratio was 2.2 in 2008 and 4.3 in 2018, with a 70% rise in acute coronary syndromes [8].

MI types 4a and 5 diagnoses, based on cardiac biomarkers changes, do not have substantial scientific evidence. Testa et al. concluded that MI diagnosis, according to the new guidelines, applies only to 15% of patients that underwent PCI, and these patients are at high risk of further adverse events during the hospital stay and at 18 months [9]. MI type 4a is related to PCI and defined as an increase in cTn values greater than five times the 99th percentile of normal baseline values or an increase greater than 20% when initial cTn values are high but stable. MI type 5 is related to CABG, defined by the elevation of cTn values greater than ten times the 99th percentile of normal baseline values.

The 2018 ESC/ACCF/AHA/WHF guideline brought some updates regarding MI diagnosis [10]. This task force considered the analytical issues of Troponin (cTn) assays, emphasized the benefits of using high-sensitivity assays, and introduced the term high-sensitivity cardiac troponin (cTn-hs) for newly available assays. Therefore, for patients with increased cTn-hs values, clinicians must distinguish between MI and non-ischemic myocardial injury. Increased cTn values greater than five times the 99th percentile of healthy subjects’ baseline values or a rise greater than 20% of cTn values define MI type 4a, and an increase in cTn values greater than ten times the 99th percentile diagnoses MI type 5.

Since 2000, the Joint European Society of Cardiology/American College of Cardiology Committee for the Redefinition of Myocardial Infarction. The assays for cTn detection have gained greater precision and sensitivity since 2000 the Joint European Society of Cardiology/American College of Cardiology Committee for the Redefinition of Myocardial Infarction [11]. Currently, some different immunoassays for cTnI detection are available. They use antibodies directed to different epitopes located on the cTn molecule. Assays antibodies attach to different epitopes and, therefore, they measure different pieces of the cTnI molecule. The issue of epitope location is essential since amino- and carboxy-terminal parts of the molecule are susceptible to proteolysis and may be related to the degree of tissue ischemia [12]. As suggested by the Committee on Standardization of Markers of Cardiac Damage of the International Federation of Clinical Chemistry and Laboratory Medicine, antibodies used for the development of reliable cTnI assays should preferably recognize epitopes that are located in the stable part of the molecule and are not affected by cTn complex and other “in vivo” modifications [13]. cTn is a complex of three proteins, present in the thin filament of the sarcomere of striated muscle, which regulates the interaction of myosin with actin in the contractile process: troponin T, which binds the complex to tropomyosin, troponin C, which binds calcium at the onset of contraction and cTnI, an inhibitor that blocks contraction in the absence of calcium [14].

The cTnI molecule has 209 amino acid residues, with a molecular weight of approximately 23–24 kDa, and three human isoforms have been described: one produced in cardiac muscle and two in skeletal muscle (“slows” TnI and “fasts” TnI). The amino acid sequence overlap between the cTnI protein and the “slows” TnI is about 40% and something close to that for the “fasts” TnI. Therefore, antibodies selected for cTnI assays should be tested to ensure they do not cross-react [15].

The recognized method for detecting most cTn is chemiluminescence. This method is a homogeneous and non-competitive immunoassay (sandwich type). The cTn value in the sample is interpreted through the light detection emitted during the chemical reaction produced by the immunocomplex. It consists of an excitation event caused by a chemical or electrochemical reaction. The light emission physical element is similar to fluorescence; it occurs from an excited state and is emitted when the electron returns to its baseline state. Chemiluminescence involves the organic compound oxidation (e.g., liminol, isoluminol, acridine, or luciterine esters) and an oxidant (e.g., hydrogen peroxide, hypochlorite, or oxygen). These catalytic reactions occur in the presence of enzymes (alkaline phosphatase, horseradish peroxidase, or micro peroxidase), metal (ionized or complexed), and hemin. These reactions range from single-step schemes, such as those involving adamantyl substrates—1,2 dioxyethane with alkaline phosphatase, to more complex multi-step reactions involving glucose-6-phosphate dehydrogenase and bacterial luciferase, coupled with nicotinamide adenine dinucleotide—flavin mononucleotide (NADH-FMN) oxidoreductase [16].

Manufacturers of the assays are responsible for preparing their standard materials through purification procedures and types of antigens (free purified or protein complexed).

Jarolin et al. proposed a simple definition based on the sensitivity range. Ultrasensitive (hs) was reserved for assays that meet specific requirements. The cTn must be measured in more than 50% of healthy subjects to label a cTn assay as highly sensitive. The 99th percentile analytical coefficient of variation (CV) values should be at most 10% [17]. Low sensitivity assays refer to the older first-generation and now outdated cTn assays. These assays detected only marked cTn increases, not small changes in cTn concentration. Apple et al. suggested the most sensitive but not ultrasensitive assays currently on the market as contemporary or medium sensitivity assays [18]. In addition to the expected loss in analytical specificity, results obtained from different analytical systems and assay generations pose a substantial problem for clinical management. Because of this scenario, in 2016, a survey was carried out at our cardiology hospital involving 202 patients [19]. They underwent PCI or CABG, and MI diagnosis confirmation was made by cardiac magnetic resonance imaging. Serum concentrations of cTnI and CKMB were evaluated before and after PCI and CABG. Of the 202 patients studied, 136 (67.3%) underwent CABG and 66 (32.7%) PCI. This study showed a lower cTn accuracy for CABG than PCI (25% and 52.4%, respectively), and CKMB accuracy was higher than cTn for CABG and PCI (76.8% and 90.5%, respectively). The best cTnI cutoff value was 5.5 ng/mL and 4.5 ng/mL for CABG and PCI, respectively. Based on these data, the authors concluded that CK-MB was more accurate than cTnI for procedure-related MI diagnoses and the need for a higher troponin cutoff value for MI types 4a and 5.

In addition, the cTn rise based on the 99th percentile leads to another possible source of disagreement between assays.

The issue of how to determine a 99th percentile value is controversial. To measure a valid 99th percentile, one must ensure that the “healthy” population is subclinical disease-free. Koerbin et al. showed that when the values were ‘coned’ by progressively excluding patients with abnormal renal function, increased NT-pro BNP, previous cardiac event, and abnormal echocardiographic, only patients <55 years old showed marked sex differences in the 99th percentile [20]. Including individuals with such comorbidities is unacceptable and will change the distribution of measured hs-cTn concentrations, substantially influencing both the length and distortion of the upper tail and significantly affecting statistical calculations [21].

Unfortunately, only some studies that standardize cTn assays can guide the physician and the scientific community in this regard. This study compared three cTnI immunoassays to identify the best cTnI cutoff values for MI type 4a and 5 diagnoses.

## 2. Materials and Methods

This study used serum samples from the MASS-V trial. The inclusion and exclusion criteria were previously described [22]. Briefly, 115 patients with multivessel stable angina symptoms with preserved left ventricular ejection fraction with a formal indication for PCI or CABG, and the criteria for MI types 4a and 5 were based on the rise of CKMB and cTnI levels but confirmed by CMR (Figure 1).

The exclusion criteria were: recent MI (less than six months), infectious or rheumatic disease, chronic renal failure (creatinine level >2.0 mg/dL), recent pulmonary embolism or venous thromboembolism (less than six months), not signing the informed consent form, contraindication for the use of glycoprotein IIb/IIIa inhibitors, and for performing magnetic resonance imaging. Blood samples were collected 6, 12, 24, 36, and 48 h after PCI and extended to 72 h in CABG. All samples were collected in a tube containing separator gel and centrifuged at 3000 ± 500 rpm for 15 min to obtain serum. Using Siemens assays, the cTnI serum concentrations were evaluated immediately after each procedure. After the first measurement, the serum samples were frozen at −80 °C for subsequent testing using other cTnI brands (Beckman and Abbott). All patients underwent CMR with the delayed gadolinium enhancement technique immediately before and after the procedure. A 1.5 Tesla magnetic resonance scanner (Philips Achieva^®^) was used, with images acquired in two long axes (2 and 4 chambers) and between eight and ten short axes of the left ventricle. The gadolinium-based contrast agent (Gadoterate meglumine Gd-DOTA, Guerbet SA^®^, France) was injected intravenously (0.1 mmol per kg of body weight), and the images were acquired after the interval of 5 to 10 min. Typical voxel size was 1.6 × 2, 1 × 8 mm, with a reconstruction matrix of 528 and reconstructed voxel size of 0.6 mm. The method for obtaining and analyzing cardiac magnetic resonance imaging was standardized in our institution. Images were analyzed by two experienced observers, with the addition of a third when consensus was not initially obtained, blinded to biochemical and surgical data. The delayed gadolinium enhancement areas were defined as an image intensity greater than two standard deviations above the mean intensities in a remote region of the myocardium on the same image and quantified with the computer-assisted planimetry program CMR42 (Circle Cardiovascular Image—Calgary—Canada).

Table 1 shows the three different kits used for cTn measurements.

Data were expressed in means ± standard deviation (SD) or median and interquartile deviation (IQR). Normality was tested using the Kolmogorov–Smirnov test. Fischer’s chi-square or exact test was used to evaluate the groups concerning their proportions. We calculated the trapezoidal area of the curves to understand their absolute quantities’ correlation better; for this correlation, the r coefficient was calculated by linear regression. The ROC curves of each assay were obtained and compared with each other, and the DeLong method was used to determine the optimal cutoff value. A logistic regression was also performed to analyze the associations. The significance level adopted for the statistical tests was 5%. The statistical software used was MedCalcversion^®^ 14.12.0 (MedCalc Software bvba, Ostend, Belgium).

## 3. Results

Patients were divided into two groups according to the procedure for which they were randomized for the MASS-V study and the Fourth Universal Definition of Myocardial Infarction classification. The distribution, demographic, and clinical data are described in Table 2.

cTnI measurements were performed in all planned samples of 115 patients using the three different brands of kits (Siemens-AdviaCentaurTnI-Ultra^®^, Beckman Coulter Access AccuTnI^®^, and Abbott Architect hs-cTnI^®^). Twenty-one (18.3%) patients had a MI confirmed by magnetic resonance imaging: 5 (5.2%) of MI type 4a, and 16 (21.3%) of MI type 5.

Figure 2 and Figure 3 show the behavior of serum cTnI levels in the three different assays. The assays performed similarly for patients submitted to CABG with peak values at 36 h. Patients submitted to PCI contemporary assays (AdviaCentaurTnI-Ultra^®^ and Beckman Coulter Access AccuTnI^®^) showed peak values also at 36 h. On the other hand, the high-sensitivity assay peak (Abbott Architect hs-cTnI^®^) was at 48 h.

We analyzed the absolute values of each peak and its rise times 99th percentile cutoff value (Table 3).

Manufacturers defined the 99th percentile of 40 ng/L for contemporary assays. For the high-sensitivity Abbott Architect hs-cTnI^®^ assay, the 99th percentile was 26.2 ng/L. The high peak of cTnI values for patients undergoing PCI was 48 h, and those submitted to CABG were 36 h. These data demonstrate that in addition to PCI’s steady rise curve, the absolute values were higher than in CABG, which differed from the Fourth Universal Definition of Myocardial Infarction. Although the distinct absolute values between the assays, they had the same behavior. Siemens assay showed an area under the trapezoidal curves of 112.5 (95%CI 71.5–231.2); Beckman assay presented an area under the trapezoidal curves of 50.23 (95%CI 41.6–75.4); and Abbott assay showed an area under the trapezoidal curves of 58.0 (95%CI 43.1–76.5). The r correlation coefficients obtained were for Siemens-AdviaCentaurTnI-Ultra^®^ vs. Beckman Coulter-Access AccuTnI^®^ (r = 0.9281; *p* < 0.0001); Siemens-AdviaCentaurTnI-Ultra^®^ vs. Abbott-Architect hs-cTnI^®^ (r = 0.8908; *p* < 0.0001); Abbott-Architect hs-cTnI^®^ vs. Beckman Coulter-Access AccuTnI^®^ (r = 0.9378; *p* < 0.0001).

Logistic regression analysis using MI type 4a and 5 as dependent variables were adjusted for diabetes, age, left ventricular ejection fraction, and previous MI. For MI type 4a, the OR for Siemens, Beckman, and Abbott was 1.62 (95%CI 1.03–2.53, *p* = 0.001), 2.16 (95%CI 1.06–4.46, *p* = 0.002) and 1.99 (95%CI 1.03–3.81, *p* = 0.002), respectively.

For MI type 5, the OR for Siemens, Beckman, and Abbott was 1.16 (95%CI 1.05–1.29, *p* = 0.004), 1.42 (95%CI 1.05–1.91, *p* = 0.023), and 1.22 (95%CI 1.01–1.47, *p* = 0.040) Siemens, Beckman, and Abbott, respectively.

Table 4 shows the analysis of ROC curves for MI type 4a, and Table 5 shows ROC curves for MI type 5.

Figure 4 shows a pairwise ROC curve comparison. For PCI, Siemens’ area under the curve (AUC) was similar to Abbott’s AUC (*p* = 0.307) but not similar to Beckman’s assay (*p* = 0.0177). However, Abbott’s AUC was similar to Beckman’s (*p* = 0.6151).

For CABG, all the AUCs were similar (Siemens vs. Beckman, *p* = 0.8079; Siemens vs. Abbot, *p* = 0.9773; Abbott vs. Beckman, *p* = 0.4683).

Low accuracy of the assays for MI types 4a and 5 diagnoses was observed when the values suggested by the current guidelines for MI were used. Among the three assays, hs-cTnI presented the worst performance. The ROC curve analysis for each assay suggested values much higher than those currently used (Table 6 and Table 7).

## 4. Discussion

Our study showed that the cutoff values for MI types 4a and 5 are much higher than that proposed by the current guidelines, regardless of the method used. For hs-cTnI, this information is already widely documented in many articles published recently, such as Omran et al. article, which found cutoff values of about 53.4% higher than that recommended by guidelines when associated with repeated revascularizations and serum concentration of more than 500 times the 99th percentile within 48 h after CABG [23]. The peak cutoff value at 48hs was greater than 13,000 ng/L (500 times more than the 99th percentile), associated with repeated revascularization. The hs-cTnI assay (Abbott—Architect STAT—High Sensitivity kit) was the same one used in our study, which showed similar results to the high-sensitivity assay. In addition to presenting a later peak for PCI (48 h later), we observed an elevation greater than 227 times the upper limit in the 99th percentile. This increase is much higher than that recommended in current guidelines for MI type 4a. Our study also showed lower accuracy for hs-cTnI compared to contemporary tests for MI types 4a and 5 diagnoses. However, after adjusting for the suggested cutoff values in our population, the performance of the assays was equivalent. According to Apple et al. [18], the most common reason for the discrepancy in hs-cTnI assays is the difference in antibody-specific epitopes. We also know that cTnI assays are influenced by multiple factors such as proteolytic degradation, phosphorylation, and complexation with other molecules (e.g., troponin C, heparin, and auto-antibodies). Different monoclonal and polyclonal antibodies are used in these assays and are probably the primary cause of their variation.

There was no consensus between the 99th percentile values of different assays. These values were obtained in diverse populations and using distinct statistical methods by the leading manufacturers. Aakre et al. [24] concluded that standardization is urgently needed and suggested that this standardization begins with a rigorous clinical and analytical screening of the individuals defined as healthy. The authors concluded that the number of individuals in the analysis, the pre-analytical, analytical, and biological factors, and the statistics might also affect the standardization (Table 8).

McKie et al. [25] suggested that myocardial imaging may differentiate between myocardial healthy and diseased, and imaging criteria have been shown to further lower the 99th percentile estimate. Therefore, it is of great value to estimate a new cutoff for diagnosing post-procedural infarction with CMR as a gold standard.

On the other hand, the Fourth Universal Definition of Myocardial Infarction established one cutoff value of cTnI for MI type 4a and 5, which does not fit with the reality of the assays. The absolute values obtained in our study were highly discrepant from the current guidelines. The trapezoidal area of the curves of each assay presented an optimal correlation with each other, showing that all the assays can be used as long as the appropriate cutoff is applied.

A well-designed study and the use of CMR for MI diagnosis before and after coronary revascularization were the strengths of our research. Additionally, the consistency of the results was observed regardless of the kit used.

This study has some limitations. First, the few samples available to perform all subsequent dosages, which impaired the full use of the MASS V study.

Second, the heterophilic antibodies presence is a concern when using immunoassays. Patients included in the study may have such antibodies. Nevertheless, we believe this fact did not influence our results because all patients presented a concentration curve rather than a plateau, which is present when dealing with interfering substances. It is also important to mention that we had the same results using assays whose antibodies were from different sources, such as polyclonal antibodies from goats and monoclonal antibodies from rats.

## 5. Conclusions

The hs-cTnI assays had lower accuracy than contemporary tests for MI types 4a and 5 diagnoses. Therefore, the cutoff values for these groups should be higher than those recommended by the manufacturers and current guidelines. The ultrasensitive and contemporary tests’ accuracy for MI types 4a and 5 diagnoses are equivalent when adjusted for these new cutoffs. It is also necessary to evaluate and validate thoroughly the analytical characteristics of the assay used, considering the loss of accuracy.

## Figures and Tables

**Figure 1 diagnostics-13-01316-f001:**
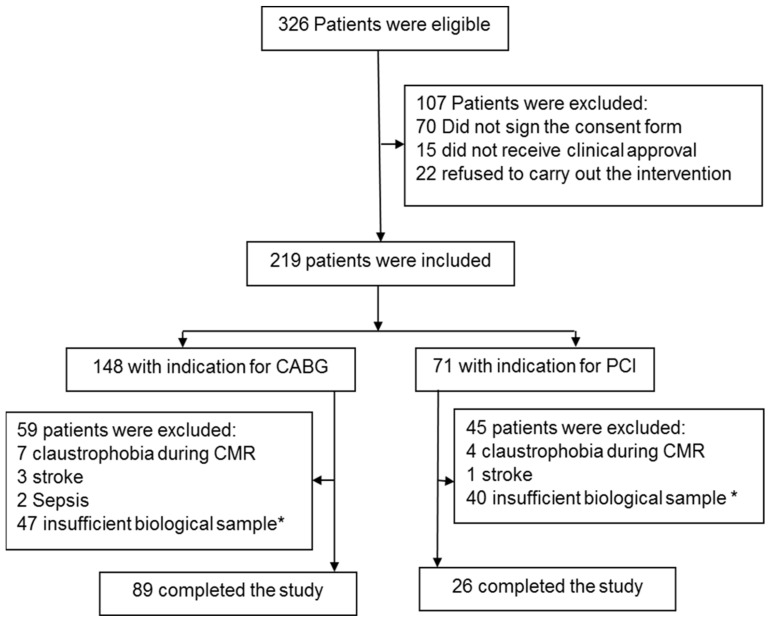
Diagram of the study. * Insufficient blood for sample analysis.

**Figure 2 diagnostics-13-01316-f002:**
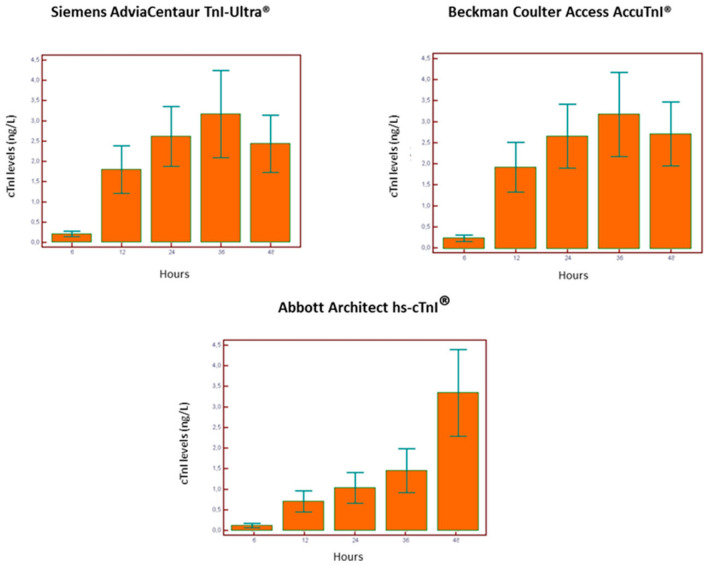
Time-related cTnI cutoff values in patients who underwent percutaneous coronary intervention.

**Figure 3 diagnostics-13-01316-f003:**
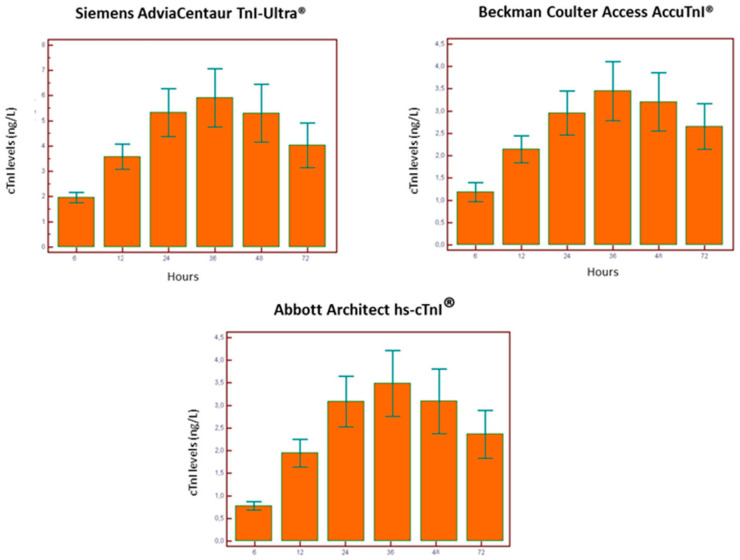
Time-related cTnI cutoff values in patients who underwent CABG.

**Figure 4 diagnostics-13-01316-f004:**
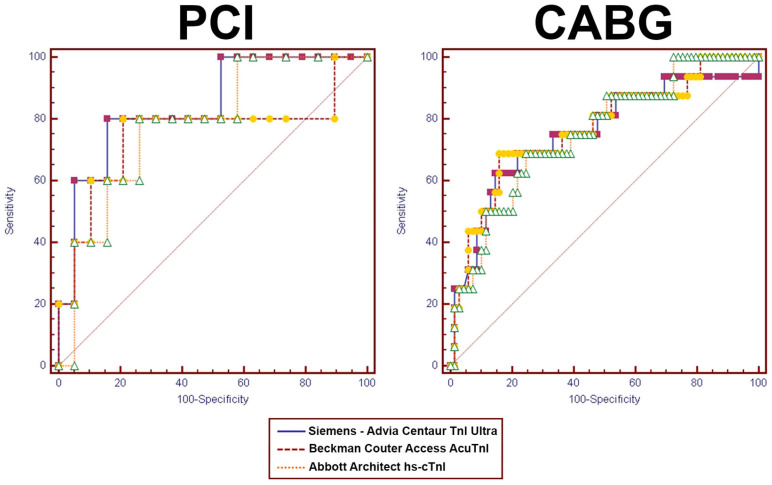
ROC Curves of cTnI in PCI and CABG.

**Table 1 diagnostics-13-01316-t001:** Biochemical and analytical performance differences between the troponin assays used in the study.

Assays	Epitopes Recognized by the Antibodies Used in the Assay	Antibody and Marker	LoB (ng\L)	LoQ (ng\L)	Cutoff at 99th Percentile (ng\L)	Percentage of Healthy Population with Detectable Values	Classification *
Siemens Advia Centaur TnI-Ultra^®^	C: 41–49, 87–91 D: 27–40	Goat polyclonal antibody and acridinium ester.	6	6	40	<32	contemporary
Beckman Coulter Access AccuTnI^®^	C: 41–49 D: 24–40	Mouse monoclonal antibody and alkaline phosphatase	1.0	2.0	40	<50	contemporary
Abbot Architect hs-cTnI^®^	C: 24–40 D: 41–49	Mouse monoclonal antibody and acridinium.	1.1–1.9	2.5	26.2	>65	Hight sensitive (hs)

* According to Pet Jerolin [17], the LoB (Blank Limit) is the highest observed measurement result of a blank sample; LoQ (Limit of Quantification): the lowest concentration detected with a total CV of 20%.

**Table 2 diagnostics-13-01316-t002:** Patient’s clinical characteristics and biochemical data.

Characteristics	Total (*n* = 115)	CABG (*n* = 89)	PCI (*n* = 26)	*p*
Demographic Profile				
Age, years	62 ± 9	62 ± 9	61 ± 9	0.7434
Female, %	38 (33%)	29 (33%)	9 (35%)	0.9655
Clinical History				
Previous MI, %	44 (38%)	29 (33%)	15 (58%)	0.0368
Diabetes mellitus, %	63 (55%)	57 (64%)	6 (23%)	0.0005
Laboratory				
Total Cholesterol, mg/dL	169 ± 49	163 ± 50	187 ± 42	0.0328
LDL cholesterol, mg/dL	98 ± 44	93 ±45	113 ± 37	0.0511
HDL cholesterol, mg/dL	40 ± 13	40 ± 12	41 ± 12	0.6910
Triglycerides, mg/dL	117 (102–132)	113 (92–128)	140 (100–170)	0.1127
Creatinine, mg/dL	1.04 (0.98–1.08)	1.06 (1.01–1.10)	0.92 (0.86–1.06)	0.0290
Reactive C-protein, mg/L	3.34 (2.65–3.63)	3.15 (2.26–3.59)	3.85 (3.08–5.60)	0.1472
Hemoglobin, g/dL	14.2 ± 1.6	14.2 ± 1.6	14.2 ±1.9	0.9117
Angiographic findings				
LVEF, %	64 ± 9	64 ± 10	62 ± 7	0.2786
All post-procedure MI, %	21(18.3%)	16 (21.3%)	5 (5.2%)	0.8863

LVEF means left ventricular ejection fraction; MI: myocardial infarction.

**Table 3 diagnostics-13-01316-t003:** cTnI rise times 99th percentile in patients submitted to PCI and CABG.

Assays	PCI ng/L	Rise Times 99th Percentile	CABG ng/L	Rise Times 99th Percentile
Siemens-AdviaCentaurTnI-Ultra^®^	2560	64 times	4830	121 times
Beckman Coulter Access AccuTnI^®^	3987	100 times	2750	69 times
Abbott Architect hs-cTnI^®^	5670	216 times	2345	90 times

CABG: coronary artery bypass graft; PCI: percutaneous coronary intervention.

**Table 4 diagnostics-13-01316-t004:** ROC Curves of cTnI assays in PCI.

Assays	Sensitivity (%)	Specificity (%)	AUC	CI 95%	Best Cutoff (ng/L)	*p*
Siemens-AdviaCentaurTnI-Ultra^®^	80	76.2	0.842	0.636–0.957	2560	0.0067
Beckman Coulter Access AccuTnI^®^	75	95	0.747	0.530–0.901	3987	0.0274
Abbott Architect hs-cTnI^®^	80	73.7	0.779	0.564 -0.921	5670	0.0345

AUC: area under the curve.

**Table 5 diagnostics-13-01316-t005:** ROC Curves of cTnI assays in CABG.

Assays	Sensitivity (%)	Specificity (%)	AUC	CI 95%	Best Cutoff (ng/L)	*p*
Siemens-AdviaCentaurTnI-Ultra^®^	62.5	86.3	0.745	0.649–0.841	4830	0.0006
Beckman Coulter Access AccuTnI^®^	68.7	64.1	0.766	0.662–0.851	2750	0.0002
Abbott Architect hs-cTnI^®^	68.7	75.4	0.753	0.647–0.840	2345	0.0001

AUC: area under the curve.

**Table 6 diagnostics-13-01316-t006:** Cutoff values and accuracy of the assays for MI type 4a.

	99th Percentile (ng/L)	Cutoff ^1^ (ng/L)	Accuracy (%)	Suggested Cutoff ^2^ (ng/L)	Corrected Accuracy (%)
Siemens	40	200	52	2560	79
Beckman	40	200	52	3987	88
Abbott	26.2	131	24	5670	87

^1^ Fourth Universal Definition of Myocardial infarction. ^2^ Adjusted according to ROC curves.

**Table 7 diagnostics-13-01316-t007:** Cutoff values and accuracy of the assays for MI type 5.

	99th Percentile (ng/L)	Cutoff ^1^ (ng/L)	Accuracy (%)	Suggested Cutoff ^2^ (ng/L)	Corrected Accuracy (%)
Siemens	40	400	31	4830	82
Beckman	40	400	26	2750	82
Abbott	26.2	261	24	2345	82

^1^ Fourth Universal Definition of Myocardial infarction. ^2^ Adjusted according to ROC curves.

**Table 8 diagnostics-13-01316-t008:** Analytical comparison of contemporary and high-sensitivity cTnI assays.

Assays *	Limits of Detection (ng/L)	99th Percentile (ng/L)/CV (%)	Minimum Value with 10% CV
Roche Elecsys	5.0	14/13	13
Abbott Architect	1.2	16/5.6	3.0
Beckman Access	2 a 3	8.6/10	3.0
Mitsubishi Pathfast	8.0	29/5	14
Nanosphere	0.2	2.8/9.5	0.5
Radiometer AQT90	9.5	23/17.5	39
Singulex Erenna	0.09	10.1/9.0	0.88
Siemens Vista	0.5	9/5.0	3
Siemens Centaur	6.0	40/10	30

SOURCE: Sherwood MV et al. High sensitivity troponin assays: Evidence. Indications and reasonable use. JAHA. 2016. CV means coefficient of variation. * Data from Apple, F.S [18].

## Data Availability

Not applicable.

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
