# Peer review of "Cardiac Troponin I in Patients Undergoing Percutaneous and Surgical Myocardial Revascularization: Comparison of Analytical Methods"

_diagnostics, 2023, doi:10.3390/diagnostics13071316_

Round 1

Reviewer 1 Report

The article submitted for review is devoted to a study on the analysis of cardiomarkers increase after percutaneous intervention and after aorto-coronary bypass surgery

Modern recommendations for the myocardial infarction diagnosis suggest its diagnosis based on an increase in the biomarkers level. However, in real clinical practice, there are a lot of reasons for increasing biomarkers and there are real difficulties in diagnosing myocardial infarction. Diagnosis of myocardial infarction of types 4A and 5 causes particular difficulties.

The article provides an overview of the available research on the study of this problem, special attention is paid to the extent to which the level of biomarkers depends on the technical side of the manufacture of reagents for the determination of biomarkers

With their research, the authors showed how often and how significantly the levels of biomarkers increase after surgical procedures when performed in chronic forms of coronary artery disease. At the same time, there is no absolute confirmation that an increase in the level of markers is associated with myocardial infarction according to MRI

Thus, the conducted study is extremely important for further revision of clinical recommendations on the diagnosis of myocardial infarction after surgical procedures

The study was conducted on a sufficient number of patients, using all the necessary research methods, a thorough statistical analysis was carried out

In the discussion, the authors conducted a review of the studies conducted on this issue, also confirming the relevance of this study.

Author Response

Dear Reviewer, thanks for your time reviewing our manuscript and comments on it. 

Reviewer 2 Report

Dear authors,
you have carried out a really interesting and clinically relevant study! Despite the restricted population (which is elaborately explained though in study flowchart and that is significant), the conclusions drawn from your study could have clinical impact once they get validated by larger studies. The study is well designed, the statistical methodology used seems robust. I have some comments that could potentially improve the manuscript:
1. My most significant concern is the non-fully addressed limitation of false positive TnI elevation which might lead to such a higher TnI cutoff. Despite the well-designed exclusion criteria and the analysis in 3 different assays, the challenging (and really frequent) probability of present heterophilic antibodies cannot be ruled out in the present study. More specifically, recent evidence suggest that the prevalence of heterophile antibodies in the population ranges from < 1% to 80%.
The main reasons for the formation of heterophile antibodies in humans are: the use of mouse monoclonal sera (antibodies) or incompletely humanized (human) antibodies for the treatment of a number of diseases (e.g., systemic connective tissue diseases or oncopathology); frequent contact with microbial antigens, animal antigens (e.g., when keeping pets), foreign proteins (e.g., in food workers, veterinarians, farmers); vaccination; blood transfusion and long-term persistence of viral agents in the body.  Serial dilution of  patients' plasma samples with control plasma (with normal troponin I levels) has led to the assumption of heterophilic antibody interference.  This assumption can be confirmed after adding heterophilic antibody blockers to the patient’s original blood sample, when the troponin I concentration is decreased (https://www.ncbi.nlm.nih.gov/pmc/articles/PMC8912997/). Hence, given the protocol of your study, I think that some false positive results cannot be ruled out and hence the specificity of your results could have been higher if you had added heterophilic antibody blockers to the patients' blood samples.

2. Please revise the title of the 7th column in table 1: "Percthe ent". Should that be "percentage"?
3. Please revise thoroughly the manuscript for errors, such as: "
Data were expressed in ± standard deviation (SD)". I guess "mean" is the word missing there (in line 127). Such errors should have not been identified by the reviewers; such errors underestimate your so well-designed study.
4. lines 186-7: please be consistent throughout the text when using commas or full stops in the numbers: "
the OR for Siemens, Beckman, and Abbott was 1.62 (95%CI 1.03-2.53, p= 0.001), 2.16 186 (95%CI 1.06-4.46, p=0,002) and 1.99 (95%CI 1.03-3.81, p=0,002), respectively." This should be: p=0.002 etc.
5. Please revise in Tables 5,6: It should be sensitivity instead of "sensibility"
6. Please provide the strengths of your study along with its limitations prior to the conclusions section.

Author Response

Dear Reviewer, thanks for your time reviewing our manuscript. Below the responses to your comments highlighted in red.

Dear authors,

you have carried out a really interesting and clinically relevant study! Despite the restricted population (which is elaborately explained though in study flowchart and that is significant), the conclusions drawn from your study could have clinical impact once they get validated by larger studies. The study is well designed, the statistical methodology used seems robust. I have some comments that could potentially improve the manuscript:

  1. My most significant concern is the non-fully addressed limitation of false positive TnI elevation which might lead to such a higher TnI cutoff. Despite the well-designed exclusion criteria and the analysis in 3 different assays, the challenging (and really frequent) probability of present heterophilic antibodies cannot be ruled out in the present study. More specifically, recent evidence suggest that the prevalence of heterophile antibodies in the population ranges from < 1% to 80%.
    The main reasons for the formation of heterophile antibodies in humans are: the use of mouse monoclonal sera (antibodies) or incompletely humanized (human) antibodies for the treatment of a number of diseases (e.g., systemic connective tissue diseases or oncopathology); frequent contact with microbial antigens, animal antigens (e.g., when keeping pets), foreign proteins (e.g., in food workers, veterinarians, farmers); vaccination; blood transfusion and long-term persistence of viral agents in the body.  Serial dilution of  patients' plasma samples with control plasma (with normal troponin I levels) has led to the assumption of heterophilic antibody interference.  This assumption can be confirmed after adding heterophilic antibody blockers to the patient’s original blood sample, when the troponin I concentration is decreased (https://www.ncbi.nlm.nih.gov/pmc/articles/PMC8912997/). Hence, given the protocol of your study, I think that some false positive results cannot be ruled out and hence the specificity of your results could have been higher if you had added heterophilic antibody blockers to the patients' blood samples.

RE: The presence of heterophilic antibodies is really a concern when using immunoassays. Nevertheless, all the patients presented a curve of concentration versus time of blood collection, rather than a plateau, present when dealing with interfering substances. The data were expressed as the mean value of all the trapezoidal curves meaning that the influence of this kind of interference should be reduced. It's important to mention we got the same results using assays whose antibodies are from different sources, showed below:

  • Siemens- polyclonal antibody from goat
  • Beckman Coulter and Abbott- monoclonal antibody from rat.

Our study's results endorse the absence of significant effects from the interfering agents.

  1. Please revise the title of the 7th column in table 1: "Percthe ent". Should that be "percentage"?

RE: Correction done.

  1. Please revise thoroughly the manuscript for errors, such as: "Data were expressed in ± standard deviation (SD)". I guess "mean" is the word missing there (in line 127). Such errors should have not been identified by the reviewers; such errors underestimate your so well-designed study.

RE: Correction done.

  1. lines 186-7: please be consistent throughout the text when using commas or full stops in the numbers: "the OR for Siemens, Beckman, and Abbott was 1.62 (95%CI 1.03-2.53, p= 0.001), 2.16 186 (95%CI 1.06-4.46, p=0,002) and 1.99 (95%CI 1.03-3.81, p=0,002), respectively." This should be: p=0.002 etc.

RE: Correction done.

  1. Please revise in Tables 5,6: It should be sensitivity instead of "sensibility"

RE: Correction done.

  1. Please provide the strengths of your study along with its limitations prior to the conclusions section.

RE: OK. Study strengths and limitations were included before the Conclusions section.

Round 2

Reviewer 2 Report

Please edit the Tables 4-5. The correct word is: "sensitivity" instead of "sensibility". Also, please comment the possibility of heterophilic antibody interference either in the Discussion or in the limitations section. This is a significant limitation despite the 3 assays utilized.

Author Response

Correction done.